# Effect of Different Forms of Selenium on the Physiological Response and the Cadmium Uptake by Rice under Cadmium Stress

**DOI:** 10.3390/ijerph17196991

**Published:** 2020-09-24

**Authors:** Haizhao Xu, Jinpeng Yan, Yan Qin, Jingmao Xu, M. J. I. Shohag, Yanyan Wei, Minghua Gu

**Affiliations:** 1Cultivation Base of Guangxi Key Laboratory for Agro-Environment and Agro-Products Safety, National Demonstration Center for Experimental Plant Science Education, College of Agriculture, Guangxi University, Nanning 530004, China; xhz2020@st.gxu.edu.cn (H.X.); 1917392033@st.gxu.edu.cn (J.Y.); qin_y@st.gxu.edu.cn (Y.Q.); xujingmao@st.gxu.edu.cn (J.X.); 2Department of Agriculture, Bangabandhu Sheikh Mujibur Rahman Science and Technology University, Gopalganj 8100, Bangladesh; islam@zju.edu.cn

**Keywords:** cadmium, selenium, rice

## Abstract

Cadmium (Cd) is a pollutant toxic to plants and a potential threat to human health. Selenium (Se), though not essential for plants, has beneficial effects on plants under abiotic stress. A hydroponic experiment was conducted to investigate the impact of different forms of Se (Nano-Se, selenite, selenate, and SeMet) on accumulation, subcellular distribution, and chemical forms of Cd, as well as oxidative stress in rice seedlings. Cd (20 μmol·L^−1^) treatment significantly decreased biomass accumulation and chlorophyll content. The application of all Se forms, except selenate, mitigated the adverse effects of Cd on growth and chlorophyll content. The application of selenite, Nano-Se, and SeMet decreased root and shoot Cd concentrations as well as root-to-shoot Cd translocation in rice seedlings. Selenate application decreased shoot Cd concentration and root-to-shoot Cd translocation with no effect on root Cd concentration. Accordingly, Se increased the sequestration of Cd in the cell wall and vacuoles and decreased the active chemical form of Cd in rice seedlings. SeMet was the most effective supplement that decreased Cd concentration and enhanced Se concentration in the roots and shoots of rice seedlings. All forms of Se further enhanced catalase (CAT) and glutathione peroxidase (GSH-Px) activities and inhibited MDA accumulation. To conclude, Se influenced Cd accumulation and translocation in rice seedlings by altering the subcellular distribution, chemical forms, and antioxidant defense system under Cd stress. These effects were highly significant with SeMet treatment, probably due to better absorption and utilization by the plant.

## 1. Introduction

Cadmium (Cd) is one of the most toxic heavy metals, and Cd soil pollution has become a serious global concern [1]. Plants grown under excess Cd show stunted growth and development due to the negative influence on physiological processes, such as photosynthesis, nutrient uptake, and water balance [2]. Moreover, a high concentration of Cd in plants causes oxidative damage via an increase in the formation of reactive oxygen species (ROS) and the effect on activities of antioxidant enzymes [3,4,5]. Rice (*Oryza sativa* L.) is one of the most important cereal crops globally, and about half of the world’s population depends on it for food. Compared to other cereals, rice is considered an efficient Cd accumulator due to its ability to absorb Cd from the soil and further translocate into the grain [6]. It is a major dietary source of Cd [7,8]. Cd concentration that exceeds the daily intake limit poses a risk to human health. Therefore, it is important to reduce the phytotoxic influence and absorption of Cd in rice to protect people from excessive Cd exposure and accumulation. 

Selenium (Se) is a micronutrient essential for humans and animals. Although Se is not essential for higher plants, several studies have demonstrated the positive effects of Se on plant growth and abiotic stress tolerance at relatively low concentrations. In recent years, increased attention has been paid to the potential role of Se in alleviating heavy metal (including Cd) toxicity in plants. Wan et al. [9] reported a decrease in root-to-shoot Cd translocation in rice seedlings supplemented with selenite [Se (IV)] and selenate [Se (VI)]. Se can combine with Cd and forms a stable Se-Cd complex in plants [10]. Ahmad et al. [11] found that Se (IV) application minimized the Cd oxidant effect and decreased lipid peroxidation and Cd uptake, transport, and distribution in *Brassica juncea* L. Additionally, Lin et al. [12] reported the ability of Se (IV) to reduce Cd availability. Se is also conducive to the recovery and reconstruction of the cell (cell membrane and chloroplast) structure under Cd stress [13]. Mozafariyan et al. [14] reported that Se (IV) addition reduced Cd accumulation and alleviated Cd toxicity in pepper via increased chlorophyll concentrations and total antioxidant activity. These studies together reveal the beneficial effects of Se, especially Se (IV) and Se (VI), in plants under Cd stress.

Different forms of Se exist in the environment. Se (IV) is the predominant form of Se in a well-drained mineral environment; however, Se (VI) is the main species in alkaline and well-oxidized conditions. Organic forms of Se also make up a major fraction of soil Se. Even low concentrations of organic Se in the soil are important as they are readily taken up by plant roots [15]. Schiavon et al. [16] found that the rate of selenomethionine (SeMet) uptake by wheat and canola was 20-fold more than that of Se (VI) or Se (IV). In recent years, nano-selenium, which is used in agricultural fields, medical therapy, as well as Se fertilization, has attracted attention [17]. Nano-sized elemental Se (selenium nanoparticles, SeNPs) formed via the reduction of Se oxyanions through biotic or abiotic pathways exist widely in the ambient [18]. Plants absorb and transform nano-selenium into inorganic Se compounds, such as Se (IV) and Se (VI), in the roots and shoots [19]. Se (IV) and Se (VI) effectively reduce Cd uptake at the whole-plant level; however, the mechanisms that alleviate Cd stress remain unaddressed. Besides, the species of Se involved in the effective mitigation of Cd phytotoxicity remain unknown.

In this study, we analyze the effects of four different forms of Se at varying concentrations on rice growth and Cd absorption and transport under Cd stress via a hydroponic experiment. This study addresses the effects of Se in alleviating Cd stress and the underlying mechanisms in rice seedlings.

## 2. Materials and Methods

### 2.1. Hydroponic Culture of Rice and Sources of Se

A hydroponic experiment was carried out at Guangxi University, Guangxi, China under the following conditions: 28/20 °C day/night temperatures, with 70% relative humidity and 240–350 μmol·m^−2^·s^−1^ light. Rice seeds (*Oryza sativa* L. cv. Y Liangyou No. 7) were obtained from the seed market in Naning, Guangxi, China. The selenite and selenate were purchased from Sigma (ST Louis, MO, USA), and the selenomethionine was purchased from Yeyuan Bio-technology (Shanghai, China). Chemosynthesized nano-selenium was prepared following the method by Wu and Ni [20]. Nano-selenium synthesized via the reduction of sodium selenite (Na_2_SeO_3_) using NH_2_OH·HCl and sodium alginate was used as the soft template. Nano-selenium particles used in this study were round or elliptic red nanospheres with a diameter of 50–100 nm. The seeds were sterilized with 35% (*v*/*v*) hydrogen peroxide (H_2_O_2_) solution for 15 min, washed three times with deionized water, placed in a plastic mesh, and soaked in 0.5 mol·L^−1^ CaSO_4_ in a container covered with aluminum foil for 10 days. After germination, uniform rice seedlings were subsequently used for the pre-hydroponic experiment in plastic pots containing 2.5 L of nutrient solution (eight plants per pot). The composition of the nutrient was (mmol·L^−1^): NH_4_NO_3_ 1.45, NaH_2_PO_4_ 0.32, K_2_SO_4_ 0.5, CaCI_2_ 1.0 mmol·L^−1^, MgSO_4_·7H_2_O 1.7, MnCl_2_·4H_2_O 9.1 × 10^−3^, (NH_4_)_6_MoO_24_·4H_2_O 5.2 × 10^−4^, H_3_BO_3_ 1.8 × 10^−2^, ZnSO_4_·7H_2_O 1.5 × 10^−4^, CuSO_4_·5H_2_O 1.6 × 10^−4^, and FeCl_2_·6H_2_O 3.6 × 10^−2^. The solution was renewed every three days throughout the entire experiment.

Fifteen days later, the seedlings were exposed to different treatments containing 20 μmol·L^−1^ CdCl_2_, with 1 μmol·L^−1^ Se as either selenite (Na_2_SeO_3_), selenate (Na_2_SeO_4_), Nano-Se or SeMet, and the treatments were represented as Cd, Cd + Nano-Se, Cd + Se (IV), Cd + Se (VI), and Cd + SeMet, respectively. A control treatment (without Cd or Se) was also included. The experiment was laid in a completely randomized design with three replicates. After 25 days, the plants were harvested, divided into shoots and roots, and washed with deionized water. The fresh weight of rice roots and shoots was recorded. Half of the rice roots and shoots were dried and weighed. Another half of the rice roots and shoots were ground into a powder in liquid nitrogen and frozen at −80 °C until further use.

### 2.2. Determination of Chlorophyll Content in Plant

The chlorophyll content was assayed spectrophotometrically. Fresh leaves (0.2 g) were added to a tube containing 2 mL of 100% acetone. After shaking for 24 h at 4 °C, the absorbance was read at 645 nm and 663 nm using a spectrophotometer to determine chlorophyll content [21].

### 2.3. Determination of Se and Cd in Root and Shoot

The root or shoot sample (0.2 g) and 6 mL of concentrated nitric acid were taken in a digestion tube, closed with the lid, and placed overnight in a fume cupboard. Subsequently, a microwave (CEM, MARS5, Charlotte, NC, USA) was used for digestion, followed by acid extraction with 6 mL of concentrated hydrochloric acid. The concentrations of Cd and Se were determined by inductively coupled plasma mass spectrometry (ICP-MS, NEXION 350X, PerkinElmer^TM^ Life Science Incorporated, Waltham, MA, USA) [9].

### 2.4. Determination of Subcellular Distribution of Cd

The subcellular components of Cd were determined by differential centrifugation [22]. The fresh root sample (0.5 g) was homogenized in 20 mL of a pre-cooled extracted solution containing 0.05 mol·L^−1^ Tris buffer (pH 7.6), 1 mmol·L^−1^ dithioerythritol, and 0.25 mol·L^−1^ sucrose. The homogenate was centrifuged for 10 min at 300 r·min^−1^ in a refrigerated centrifuge and precipitated to obtain the cell wall. The supernatant was subsequently centrifuged for 15 min at 2000 r·min^−1^ and precipitated to obtain the organelles (nucleus and chloroplast). The supernatant was further centrifuged for 20 min at 10,000 r·min^−1^ and precipitated to obtain the cell membranes. The final supernatant is the soluble fraction of the cell that contains vacuoles, ribosomes, and nuclear proteins. All operations were performed at 4 °C.

### 2.5. Determination of Chemical Forms of Cd

Cadmium chemical forms were extracted and separated following the sequential extraction method [23]. The extraction agents (1) 80% ethanol (extraction of inorganic salts, mainly nitrates and chlorides, as well as amino acid salts); (2) deionized water (H_2_O, extraction of organic salts); (3) 1 mol·L^−1^ sodium chloride (NaCl, extraction of pectin salt, which extracts the heavy metal in the binding state or adsorption state with protein); (4) 2% acetic acid (HAc, extraction of insoluble heavy metal phosphates); and (5) 0.6 mol·L^−1^ hydrochloric acid (HCl, extraction of oxalates) were used. The concentration of Cd obtained in each extraction step was measured using an inductively coupled plasma-mass spectrometer (ICP-MS, NEXION 350X, PerkinElmerTM Life Science Incorporated, Waltham, MA, USA)

### 2.6. Determination of Malondialdehyde, Superoxide Dismutase, Peroxidase, Catalase, and Glutathione Peroxidase 

To determine the antioxidant enzyme activity, fresh root (0.2–0.5 g) was rinsed; the root sample and PBS were taken in a centrifuge tube at a ratio of 1:4 (weight: volume; g: mL). Approximately 20% of the tissue homogenate was ground with a tissue tamper (10,000–15,000 r·min^−1^) and centrifuged at 4000 r·min^−1^ at 4 °C for 10 min. The resulting supernatant was collected to determine superoxide dismutase (SOD), catalase (CAT), peroxidase (POD), and glutathione peroxidase (GSH-Px) activities and malondialdehyde (MDA) content according to the method by Li et al. [24] using the assay kits (Jiancheng Bioengineering Institutes, Nanjing, China).

### 2.7. Statistical Analyses

All data were expressed as mean ± standard deviation of three replicates. Statistical analysis software SPSS19.0 (International Business Machines Corporation, New York, NY, USA) was used to test the significance of the difference at 95% confidence interval. The data were analyzed and plotted using GraphPad Prism 7 (GraphPad Software Inc., San Diego, CA, USA).

## 3. Results

### 3.1. Effects of Se Forms on Growth and Chlorophyll Content of Rice under Cd Stress

Compared to the control, the root and shoot dry weights of rice seedlings treated with Cd were significantly reduced by 23% and 30%, respectively (Table 1). However, when Cd-stressed plants were supplemented with SeMet, Se (IV), and Nano-Se, the root dry weight significantly increased by 118%, 67%, and 30%, respectively, compared with Cd alone treatment. The shoot dry weight of rice seedlings treated with SeMet and Se (IV) significantly increased by 51% and 96%, respectively, compared with the seedlings treated with Cd alone. Meanwhile, Se (VI) treatment significantly decreased the root dry weight of rice seedlings by 11%.

Chlorophyll content in rice leaves was significantly reduced by 29% under Cd stress, compared with the control treatment (Table 1). However, the chlorophyll content of Cd + SeMet treatment was significantly increased by 18%, compared with Cd alone treatment. There was no significant (*p* > 0.05) difference in chlorophyll content among Cd + Nano-Se treatment, Cd + Se (IV) treatment, Cd + Se (VI) treatment, and Cd alone treatment. SeMet supplement was more effective in improving root and shoot dry weights of rice seedlings and chlorophyll content under Cd stress than other forms of Se.

### 3.2. Effects of Se Forms on Cd Content in Rice Roots and Shoots and Transport Factor

Cd concentration in rice roots and shoots were affected by different levels and forms of Se under Cd stress (Figure 1). Cd + Nano-Se treatment, Cd + Se (IV) treatment, and Cd + SeMet treatment significantly reduced Cd content in the roots by 11%, 5%, and 16%, respectively, compared with Cd alone treatment (Figure 1A). However, there was no significant difference in root Cd content between Cd + Se (VI) treatment and Cd alone treatment. Cd content in the shoots was significantly reduced by all forms of Se (Figure 1B); Cd + SeMet treatment showed maximum reduction (43%). Figure 1C shows the transfer factor of Cd from roots to shoots for rice seedlings supplemented with different forms of Se. The transfer factor of Cd alone treatment (0.18) was significantly more than that of all forms of Se supplements. The transfer factor of Cd + Nano-Se treatment was significantly more than that of the other three Se treatments. The significant changes in transfer factor and shoot Cd content suggest that Se treatment can reduce root-to-shoot Cd translocation.

### 3.3. Effects of Se Forms on Subcellular Distribution of Cd in Rice Seedlings under Cd Stress

Differential centrifugation was used to separate plant cell components into four parts: cell wall, organelles, cell membrane, and soluble fraction. Significant differences were observed in the proportion of Cd concentration among the different fractions (Figure 2A). In the roots, Cd in Cd alone treatment was mainly associated with the cell wall (50%) and the soluble fraction (41%) and a minor part (9%) was stored in the organelles and the cell membranes. Meanwhile, in Se treatment (any form), most of the Cd was present in the soluble fraction (48%–58%). The proportion of Cd in the soluble fraction increased with Se supplements, of which the proportion in Cd + SeMet treatment was 17% higher than that in Cd alone treatment.

The subcellular distribution of Cd in the shoots showed a trend similar to that in the roots (Figure 2B). The proportion of Cd in the cell wall was highest in the seedlings treated with Cd alone, which was significantly higher than that of Cd + Nano-Se treatment, Cd + Se (IV) treatment, and Cd + SeMet treatment. The proportion in the soluble fraction in all forms of Se was also more than that in Cd alone treatment. However, for roots and shoots, there was no significant (*p* > 0.05) difference in the proportion of Cd in the organelles and cell membranes between Cd alone treatment and Se treatments.

### 3.4. Effects of Se Forms on the Chemical Forms of Cd in Rice Seedlings under Cd Stress

The chemical forms of Cd in roots are shown in Figure 3A; the forms of Cd extracted by 2% HAc and 1 M NaCl were dominant in each treatment in the roots (42.5%–54% of total Cd). Meanwhile, the fraction extracted by 0.6 M HCl was low. In the Cd alone treatment, water-soluble Cd and Cd extracted by 80% ethanol accounted for 35.5% of total Cd in roots. Higher amounts of Cd extracted by 80% ethanol and water were observed in Cd alone treatment than that in Se treatments. Whereas Cd extracted by 0.6 M HCl in Cd + Se (VI) treatment was the highest. In the shoots, the trend was similar to that in the roots (Figure 3B). Cd fractions extracted by 1 M NaCl and 2% HAC were also predominant in the shoots, accounting for 54%–60% of total Cd. However, Cd extracted by 1 M NaCl, 80% ethanol, and 0.6 M HCl were higher in all forms of Se treatments than that in Cd alone treatment, while Cd extracted by water was lower than that in Cd alone treatment.

### 3.5. Effects of Se Forms on Se Concentration in Roots and Shoots of Rice Seedlings under Cd Stress

The concentration of Se in the roots and shoots of rice seedlings under Cd stress was significantly affected by Se supplements (Figure 4). No significant differences were observed in the root and shoot Se between Cd alone and control. Compared to the Cd alone treatment, Cd + Nano-Se, Cd + Se (IV), Cd + Se (VI), and Cd + SeMet treatments significantly increased Se in the roots by 2.8, 15.7, 1.6, and 34.0 fold, respectively, and in the shoots by 0.7, 5.9, 1.0, and 99.5 fold, respectively.

### 3.6. Effects of Se Forms on Antioxidant Enzyme Activities in Rice Seedlings under Cd Stress

We further investigated the effects of different forms and concentrations of Se on antioxidant enzyme activities in rice seedlings under Cd stress. In the Cd alone treatment, MDA content was significantly increased by 372% compared with that in control (Figure 5A). Moreover, POD and SOD activities increased significantly by 129% and 107%, respectively, under Cd stress compared with the control (Figure 5B,C), while CAT and GSH-Px activities significantly decreased by 73% and 32%, respectively (Figure 5D,E). All Se supplements significantly reduced the increase in Cd-mediated MDA content, but did not return to the level of control. The MDA levels of Cd + Nano-Se treatment, Cd + Se (VI) treatment, Cd + Se (IV) treatment, and Cd + SeMet treatment decreased by 18%, 32%, 39%, and 52%, respectively, compared with Cd alone treatment. Se addition decreased the POD and SOD activities to different levels, compared with Cd alone treatment. Among them, POD activities of the leaves treated with Nano-Se, Se (VI), Se (IV), and SeMet significantly decreased by 20%, 11%, 32%, and 42%, respectively, compared to the Cd alone treatment. The SOD activities in the leaves were reduced by 32%, 29%, 42%, and 48% in Cd + Nano -Se treatment, Cd + Se (VI) treatment, Cd + Se (IV) treatment, and Cd + SeMet treatment, respectively, and the SOD activity in Cd + SeMet treatment was close to that in control. In terms of CAT activity, Se addition, especially Cd + SeMet treatment (increased by 2.05-fold compared with the Cd alone treatment), effectively alleviated Cd-induced reduction in CAT activity. Additionally, CAT activities in Cd + Nano-Se treatment, Cd + Se (VI) treatment, and Cd + Se (IV) treatment were significantly increased by 0.56, 1.00, and 1.45 fold, respectively, compared with the Cd alone treatment. A similar trend was observed in GSH-Px activity. Se addition significantly increased the GSH-Px activity compared to Cd alone treatment. Additionally, compared to the control treatment, the GSH-Px activities in Cd + Se (IV) treatment and Cd + SeMet treatment were significantly increased by 16% and 97%, respectively.

## 4. Discussion

Cd contamination in soil has become a global concern. Several studies have demonstrated the harmful effects of Cd on plant growth, mainly manifested as leaf chlorosis, restricted root development, and decreased biomass [25,26]. In our study, rice seedlings grown in a solution containing 20 μM of Cd alone for 15 days showed stunted growth (Table 1). The reduction in biomass accumulation (mainly fresh weight) under Cd stress may be due to chlorophyll degradation in rice seedlings (Table 1). The observed degradation of chlorophyll in rice seedlings grown under Cd stress is consistent with the findings of Tang et al. [27]. They found that exposure to Cd stress diminished the chlorophyll content by about 27.5% compared to control. Cd stress, which affects the structure of chloroplast and chloroplast membrane, promotes ion exchange in the chloroplasts and inhibits the essential enzymes of the Calvin cycle, reduces chlorophyll content, and ultimately inhibits plant growth [28]. Although Se is not an essential element for plants, studies have proven its ability to improve plant growth under heavy metal stress [13,27]. In our study, Se promoted biomass accumulation, and SeMet resulted in a pronounced effect. The positive impact of Se on the growth of rice seedlings under Cd stress may be due to the reversal of Cd-induced chlorophyll degradation (Table 1). Filek et al. [29] confirmed less degradation of chloroplasts in plants cultured in media containing Se under Cd stress. Our findings are consistent with this earlier report. Similarly, Ahmad et al. [11] reported that Se protects plants under stress via enhanced starch accumulation in the chloroplasts. 

The heavy metal Cd is easily transferred from soil to plants under Cd stress. Studies have reported that the use of inorganic Se decreased Cd uptake in tomato [21], pakchoi [30], wheat [31], and other plants [10,32]. In this study, the addition of Nano-Se, Se (IV), and SeMet significantly reduced the Cd content in rice tissues (shoots and roots), and this effect was pronounced with the SeMet application. Contrarily, a considerable variation in root Cd content was not found with the addition of Se (VI), while a decline in shoot Cd content was observed (Figure 1). Similarly, Yu et al. [30] reported that Se (IV) reduced Cd content in the shoots of pakchoi, while Se (VI) increased. Alves et al. [21] reported that both Se (IV) and Se (VI) reduced Cd uptake in tomato (*Solanum lycopersicum* L.). These results suggest that the specific effect depends on Cd dosage and plant species as well as Se species. Moreover, all species of Se reduced the transfer factor of Cd in rice seedlings (Figure 1). Rice roots act as a barrier against Cd translocation, while the beneficial effect of Se was significantly related to the reduction in Cd uptake or translocation towards the shoots.

In roots, the accumulation and proportion of Cd in the different subcellular fractions were as follows: cell wall (50%) > cytosol (41%) > organelles (9%) (Figure 2). This pattern could be related to the specific mechanisms employed by the plant to alleviate Cd stress. In plants, the cell wall is the first barrier, which prevents Cd ions from entering into the plant [33]. Due to the limited ability of the cell wall to fix Cd at high concentrations, the soluble fraction merges Cd with thiol (–SH) groups (e.g., GSH and phytochelatins [PCs]) to form complexes, which induce less cell organelle damage [34]. Schmöger et al. [34] observed that sulfur can promote vacuole compartmentalization of Cd. Similar mechanism with Se. Exogenous Se could significantly reduce the proportion of Cd in the cell wall and subsequently increase the proportion of Cd in the soluble fraction (Figure 2); this finding is consistent with the reports of Yu et al. [30]. These results suggest that Se can promote vacuole and cell wall sequestration of Cd in the roots. Se, which is chemically similar to sulfur, may also contribute to Cd sequestration by stimulating the production of GSH and PCs in plants [35]. Moreover, in plants, Cd is absorbed by the roots and subsequently transferred to the shoots. The forms of Cd that already exist in plants can directly affect the migration ability and activity, and Cd fractions extracted by 80% ethanol and water are the most active [23]. In this study, the proportions of Cd extracted by 2% HAc and 1 M NaCl were dominant in the plant, similar to the results of Wang et al. [36]. The addition of Se markedly reduced the proportions of Cd extracted by 80% ethanol and water in the roots (Figure 3). These results indicate that Se reduces Cd accumulation in shoots by inhibiting root-to-shoot Cd translocation. To conclude, Se increases sequestration of Cd in cell walls and vacuoles and decreases the active chemical form of Cd in rice seedlings, thereby reducing Cd accumulation in the shoots

Differences in the effects of Se species on the concentration and distribution of Cd in rice seedlings were probably due to the differences in their absorption and assimilation (Figure 4). Rice plants preferentially take up and accumulate organic forms of Se compared to inorganic forms [15,32,37,38]. Moreover, organic Se is easier to transport to the shoot [39]. Previous studies have found that Nano-Se is absorbed by plant roots through physical or chemical pathways, and small Nano-Se forms easily pass through the cell wall barrier and get absorbed by the plant roots [40], therefore cell wall is the primary limiting factor for Nano-Se absorption. In summary, the study reveals a better effect of SeMet due to improved plant absorption and utilization.

Heavy metals exert toxic effects on plants through the generation of free radicals, such as superoxide radical (O^2−^), hydrogen peroxide (H_2_O_2_), and hydroxyl radical (OH), called reactive oxygen species (ROS). ROS initiate the peroxidation and destruction of the lipid bilayer that lead to irreparable metabolic dysfunction and cell death [21]. In our study, Cd increased the MDA level in rice roots (Figure 5), probably due to increase lipid peroxidation caused by Cd-induced oxidative stress. Similar findings have been reported in rape and wheat seedlings [41]. Under such stressful situations, plants develop an antioxidant defense system, which plays a vital role in alleviating oxidative stress. In the current study, the activities of CAT and GSH-Px considerably reduced while that of SOD and POD significantly increased under Cd stress (Figure 5). Antioxidant enzymes exhibit dual behavior under Cd stress depending on Cd dose and plant species. Tang et al. [42] reported POD and SOD activities of rice seedlings were significantly increased under Cd stress. Maksymiec et al. [43] found that Cd at high concentrations increased free radical production, enhanced SOD activity, and decreased CAT activity in *Arabidopsis thaliana*. The decline in CAT and GSH-Px activities in plants under Cd stress suggests the decrease in antioxidant efficiency. However, the decrease in MDA level is due to the reduction of Cd-induced oxidative stress by exogenous Se. A sharp increase in CAT and GSH-Px activities and a decrease in SOD and POD activities in shoots as compared to Cd alone treated seedlings were observed in this study (Figure 5). Thus, Se helps rice seedlings to adapt to Cd stress. Similarly, Se regulated antioxidant enzyme activities (e.g., CAT, GSH-Px, SOD, and POD) in plants under heavy metal stress [26,27]. In our study, the effect of Se in alleviating Cd stress was dependent on the Se species. SeMet application was the most effective in regulating the antioxidant enzyme activities.

## 5. Conclusions 

To conclude, this study reveals that the effects of Se on the physiological response and Cd uptake of rice seedlings depend on the form of Se applied. All forms of Se, except Se (VI), significantly enhanced biomass accumulation and chlorophyll content. Besides, Se application affected Cd accumulation in rice seedlings by altering Cd subcellular distribution (in cell walls and vacuoles) and forms of 2% HAc and 1 M NaCl extracted Cd with low mobility, and by enhancing antioxidative defense. However, such effects depend on the Se form applied. Among the different forms of Se used in this study, SeMet resulted in a pronounced effect in alleviating Cd stress in rice seedlings.

## Figures and Tables

**Figure 1 ijerph-17-06991-f001:**
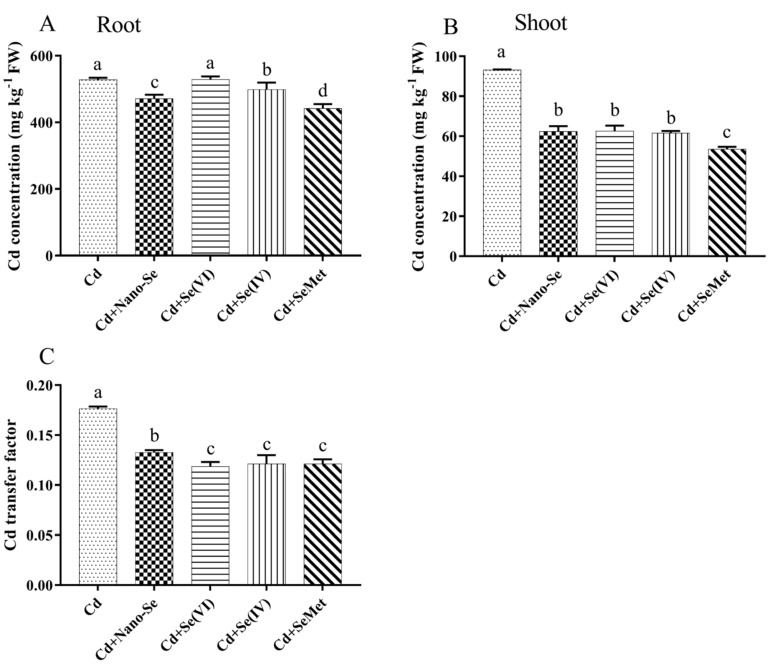
Effects of different Se forms on the Cd concentration in the roots (**A**) and shoots (**B**) and Cd transfer factor (**C**) in rice seedlings in the presence of Cd. Data points and error bars represent mean values ±SD (*n* = 3). The different letters indicate statistically significant differences between the treatments at *p* < 0.05.

**Figure 2 ijerph-17-06991-f002:**
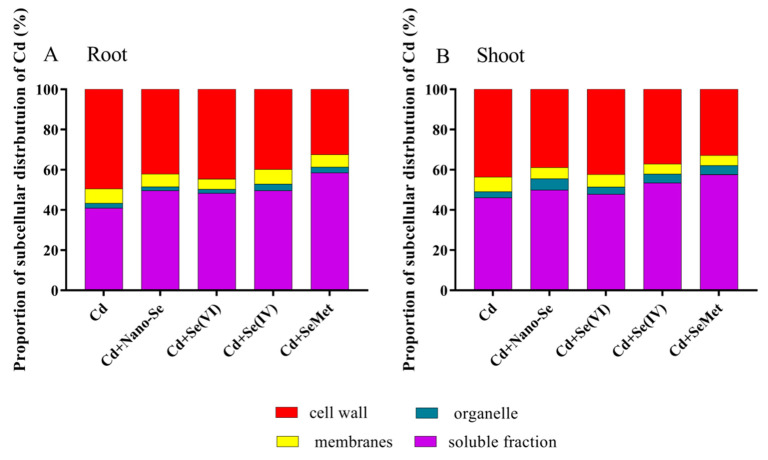
Effects of different Se forms on subcellular distribution of Cd in the roots (**A**) and shoots (**B**) of rice seedlings in the presence of Cd.

**Figure 3 ijerph-17-06991-f003:**
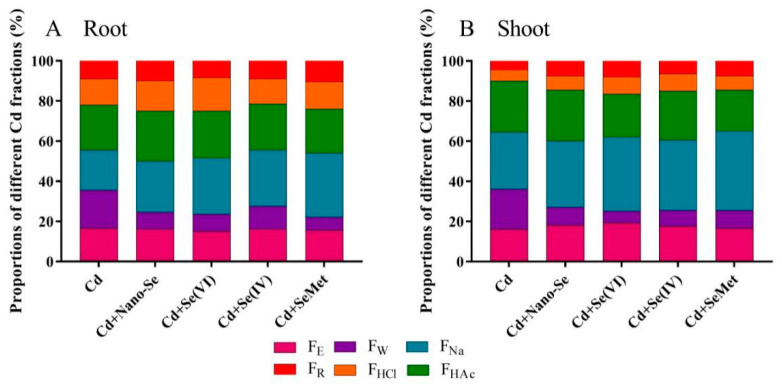
Effects of different Se forms on chemical forms of Cd in the roots (**A**) and shoots (**B**) of rice seedlings in the presence of Cd.

**Figure 4 ijerph-17-06991-f004:**
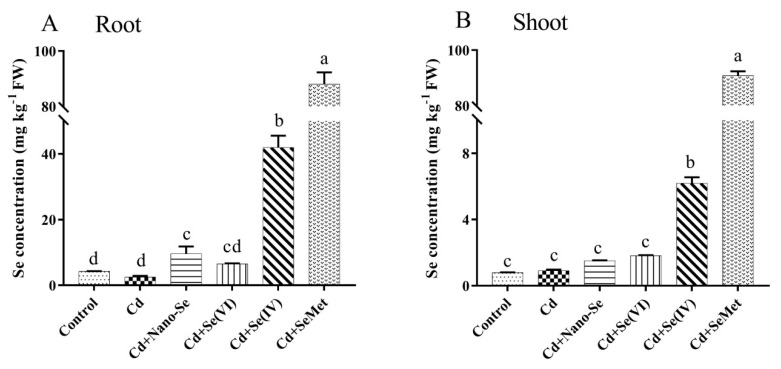
Effects of different Se forms on Se concentration in the roots (**A**) and shoots (**B**) of rice seedlings in the presence of Cd. Data points and error bars represent mean values ±SD (*n* = 3). The different letters indicate statistically significant differences between the treatments at *p* < 0.05.

**Figure 5 ijerph-17-06991-f005:**
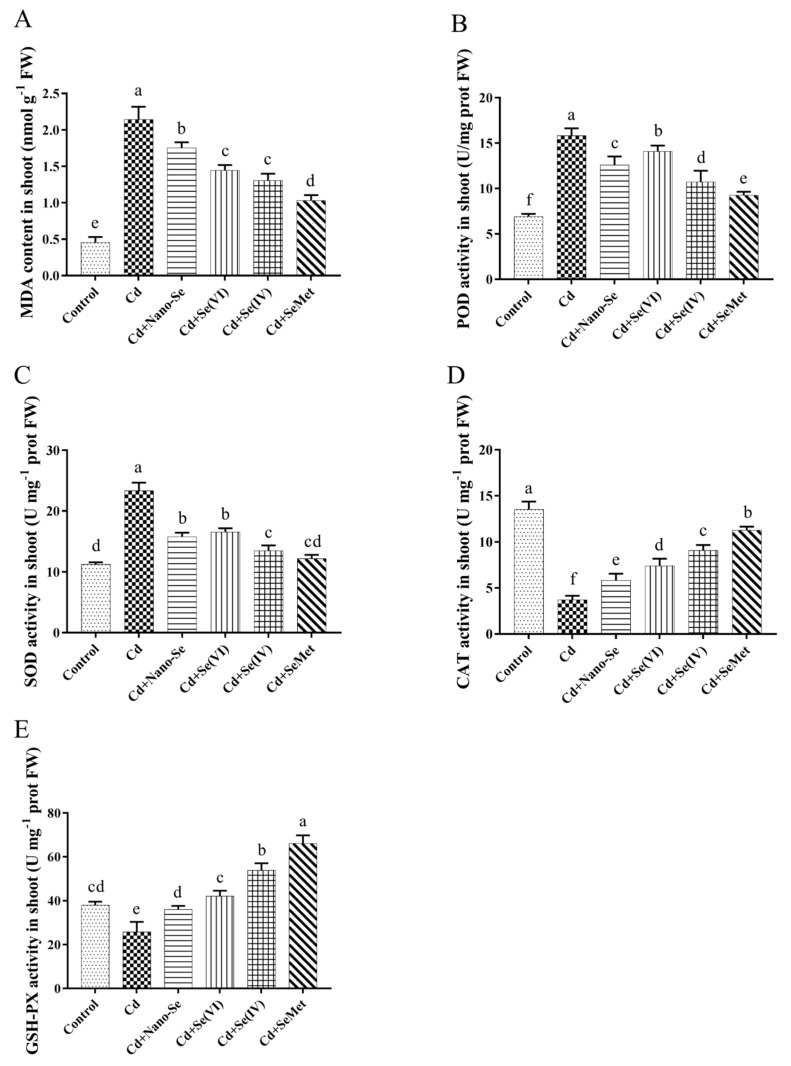
Effects of different Se forms on malondialdehyde (MDA) content (**A**) and peroxidase (POD) (**B**), superoxide dismutase (SOD) (**C**), catalase (CAT) (**D**), and glutathione peroxidase (GSH-Px) (**E**) activities of rice seedlings in the presence of Cd. Data points and error bars represent mean values ±SD (*n* = 3). The different letters indicate statistically significant differences between the treatments at *p* < 0.05.

**Table 1 ijerph-17-06991-t001:** Effects of Se forms on the mass of roots or shoots and chlorophyll content in the presence of Cd. Data are expressed as mean values ±SD (*n* = 3). The different letters indicate statistically significant differences between the treatments at *p* < 0.05.

Treatments	Root Mass (mg·plant^−1^)	Shoot Mass (mg·plant^−1^)	Chlorophyll Content (mg·g^−1^)
Control	118.77 ± 1.10 ^c^	687.50 ± 15.00 ^c^	4.26 ± 0.10 ^a^
Cd	91.06 ± 2.32 ^d^	484.00 ± 8.72 ^d^	3.30 ± 0.10 ^c^
Cd + Nano-Se	118.29 ± 1.21 ^c^	465.83 ± 21.26 ^d^	3.44 ± 0.10 ^c^
Cd + Se (VI)	80.98 ± 3.23 ^e^	312.50 ± 12.50 ^e^	3.35 ± 0.07 ^c^
Cd + Se (IV)	152.08 ± 1.40 ^b^	948.89 ± 20.09 ^a^	3.52 ± 0.30 ^c^
Cd + SeMet	198.77 ± 1.52 ^a^	731.11 ± 8.39 ^b^	3.89 ± 0.07 ^b^

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
