# Peer review of "Effect of Different Forms of Selenium on the Physiological Response and the Cadmium Uptake by Rice under Cadmium Stress"

_ijerph, 2020, doi:10.3390/ijerph17196991_

Round 1

Reviewer 1 Report

Minor Comments

1.The authors should indicate where they obtained selenium derivatives, and as for selenium nanoparticles what method was used for their production and what was the size of Se nanoparticles

2.line 41 - use Italic for ‘Orisa sativa’;  and the same for the references: 2, 4, 5, 10, 11, 19, 25, 26, 27, 29, 32, 35, 42

3.Abstract: ‘Cd (20 μmol L−1) treatment significantly  increased biomass accumulation and chlorophyll content.’- do you mean ‘increase’ but not ‘decrease’?

4.Line 160  ‘Chlorophyll content in rice leaves significantly (p  < 0.05) reduced by 29% under Cd stress’ should be changed to ‘…was reduced….’

5.Line 161 ‘However, rice under Cd stress supplemented with  SeMet was significantly  (p  < 0.05)  increased  18%,  compared to Cd alone treatment”.- grammar; Are you speaking about chlorophyll or rice?

  1. ‘Table 1. Effect of Se supply in the dry weight of root or shoot and chlorophyll content’- change to ‘on dry weight’
  2. 7. Table 1- will it be better to write ‘root/shoot mass’ but not ‘root/shoot dry weight’?- as ‘dry weight’ is a special indicator of plant physiology and is expressed in %.
  3. Figure 2: delete ‘Data points and error bars represent mean values±SD (n=3). The different letters indicate statistically significant differences between the treatments at the P<0.05’. The same for Figure 3
  4. 9. It is desirable to use both ‘A’, ‘B’ and ‘Root/Shoot’ indications in all Figures
  5. Line 235; ‘MDA significantly (p < 0.05)  increased’ change to: ‘MDA content significantly (p  <  0.05)  increased

11 line 244: ‘Among them, the leaves POD activities treat with Nano-Se, Se(VI), Se(IV), and 

SeMet significantly…’ change to : ‘Among them, POD activities of leaves treated with Nano-Se, Se(VI), Se(IV), and  SeMet significantl’…

  1. 12. Line 246 :’ The SOD activity in the leaves reduce.’ change to: ‘The SOD activity in the leaves reduced by’…

13.line 248: ‘In  terms of CAT activity,  Se addition were effective in alleviating Cd-inhibited leaf CAT activity’ change to ‘In  terms of CAT activity,  Se addition was effective in alleviating Cd-inhibited leaf CAT activity’

  1. line 253: ‘the GSH-Px active center is selenocysteine, so selenium is an essential part of GSH-Px.’. As far as I know there are no data proving the existence of Se in the active center of GSH-Px in plants, only in mammals. If the authors have appropriate information, please add a reference
  2. 15. line 255 Please check the sentence: ‘Additionally, Cd + Se(IV) treatment and Cd + SeMet treatment significantly (p < 0.05) 16% and 97% higher than the control treatment.’- there is no predicate and it is not clear what the authors are speaking about
  3. 16. Line 286: add predicate to the sentence: ‘Similarly, Yu et al. [29] that  Se(IV) reduced Cd content in the shoots of pakchoi while Se(VI) increased’-
  4. 17. Line 300: ‘Exogenous Se could significantly (p < 0.05)  reduced Cd in  the cell  wall and subsequently increased the proportion of Cd in soluble fractions’- change to ‘Exogenous Se  could significantly  (p < 0.05)  reduce Cd in  the cell  wall and subsequently increase the proportion of Cd in soluble fractions

Author Response

We are highly grateful for the helpful comments from the editor and referees that have been considered and carefully incorporated into the revised version of the manuscript. Please see our point-by-point response to the editor/reviewer’s comments.

Response to Reviewers’ Comments

Reviewer #1:

Minor Comments

  1. The authors should indicate where they obtained selenium derivatives, and as for selenium nanoparticles what method was used for their production and what was the size of Se nanoparticles.

Thank you very much for this valuable advice. Chemosynthesized nano-selenium was prepared following the method by Wu and Ni [20]. Nano-selenium synthesized via the reduction of sodium selenite (Na2SeO3) using NH2OH·HCl and sodium alginate was used as the soft template. Nano-selenium particles used in this study were round or elliptic red nanospheres with a diameter of 50–100 nm. We have modified the methods accordingly to address the reviewer’s comment.

  1. line 41 - use Italic for ‘Orisa sativa’; and the same for the references: 2, 4, 5, 10, 11, 19, 25, 26, 27, 29, 32, 35, 42

Thank you and, we have modified the scientific name as ‘Oryza sativa’ in the revised version of the manuscript.

  1. Abstract: ‘Cd (20 μmol L−1) treatment significantly increased biomass accumulation and chlorophyll content.’- do you mean ‘increase’ but not ‘decrease’?

Thank you very much for this helpful question. We have re-checked the data and realized that this was an error that occurred during the writing process. We truly apologize for this. We have accordingly revised the sentence to “Cd (20 μmol L−1) treatment significantly decreased biomass accumulation and chlorophyll content”. Please find the modification in the revised version of the manuscript (lines 20–21)

  1. Line 160 Chlorophyll content in rice leaves significantly (p < 0.05) reduced by 29% under Cd stress’ should be changed to ‘…was reduced….’

We have corrected this error in the revised version of the manuscript (line 165).

  1. Line 161 ‘However, rice under Cd stress supplemented with SeMet was significantly (p < 0.05) increased 18%, compared to Cd alone treatment”. grammar; Are you speaking about chlorophyll or rice?

Thank you for the question. We have modified the sentence as follows: “However, the chlorophyll content of Cd + SeMet treatment was significantly increased by 18%, compared with Cd alone treatment.” Please find the revision in the revised version of the manuscript (lines 166–167).

  1. ‘Table 1. Effect of Se supply in the dry weight of root or shoot and chlorophyll content’- change to ‘on dry weight’

Thank you. The sentence has been revised (lines 167-168).

  1. Table 1- will it be better to write ‘root/shoot mass’ but not ‘root/shoot dry weight’?- as ‘dry weight’ is a special indicator of plant physiology and is expressed in %.

Thank you very much for your valuable suggestion. The heading of Table 1 has been revised.

  1. Figure 2: delete ‘Data points and error bars represent mean values ± SD (n=3). The different letters indicate statistically significant differences between the treatments at the P<0.05’. The same for Figure 3

Thank you very much for this comment. Accordingly, we have deleted the sentences from the Figure legends.

  1. It is desirable to use both ‘A’, ‘B’ and ‘Root/Shoot’ indications in all Figures

Thank you very much for this suggestion. Accordingly, we have specified the root and shoot in the figure legend. Please find the revision in revised the Figures.

  1. Line 235; ‘MDA significantly (p < 0.05) increased’ change to: ‘MDA content significantly (p < 0.05) increased

Thank you. Accordingly, we have revised “MDA” to “MDA content”. Please find the revision in the revised version of the manuscript (line 234).

  1. line 244: ‘Among them, the leaves POD activities treat with Nano-Se, Se(VI), Se(IV), and SeMet significantly…’ change to : ‘Among them, POD activities of leaves treated with Nano-Se, Se(VI), Se(IV), and SeMet significantl’…

Thank you very much. The sentence has been revised accordingly (line 243).

  1. Line 246 :’ The SOD activity in the leaves ’ change to: ‘The SOD activity in the leaves reduced by’…

Thank you. We have revised the sentence accordingly. Please find this change in the revised version of the manuscript (line 245).

  1. line 248: ‘In terms of CAT activity, Se addition were effective in alleviating Cd-inhibited leaf CAT activity’ change to ‘In terms of CAT activity, Se addition was effective in alleviating Cd-inhibited leaf CAT activity’

Thank you. We have revised the sentence accordingly (lines 247-249).

  1. line 253: ‘the GSH-Px active center is selenocysteine, so selenium is an essential part of GSH-Px.’. As far as I know there are no data proving the existence of Se in the active center of GSH-Px in plants, only in mammals. If the authors have appropriate information, please add a reference

Thank you for this valuable comment. The sentence has been deleted. We sincerely apologize for this error. 

  1. line 255 Please check the sentence: ‘Additionally, Cd + Se(IV) treatment and Cd + SeMet treatment significantly (p < 0.05) 16% and 97% higher than the control treatment.’- there is no predicate and it is not clear what the authors are speaking about

The sentence has been revised as “Additionally, compared to the control treatment, the GSH-Px activities in Cd + Se(IV) treatment and Cd + SeMet treatment were significantly increased by 16% and 97%, respectively.” Please find the revision in the revised version of the manuscript (lines 249–252).

  1. Line 286: add predicate to the sentence: ‘Similarly, Yu et al. [29] that Se(IV) reduced Cd content in the shoots of pak choi while Se(VI) increased’-

We have made the necessary correction (lines 285–286).

  1. Line 300: ‘Exogenous Se could significantly (p < 0.05) reduced Cd in the cell wall and subsequently increased the proportion of Cd in soluble fractions’- change to ‘Exogenous Se could significantly (p < 0.05) reduce Cd in the cell wall and subsequently increase the proportion of Cd in soluble fractions

Thank you for this suggestion. Accordingly, we have revised the sentence (lines 299- 300).

We hope that the reviewer finds the revision satisfactory.

Reviewer 2 Report

Xu et al manuscript entitled as “Effect of different forms of selenium on the physiological response and the cadmium uptake by rice under cadmium stress” studied the effect different form of Se on Cd stressed conditions in hydroponically grown rice seedlings. And authors concluded Se Met shown efficient effect on alleviating Cd stress in rice seedling than that other forms of Se.

Please find the some of the queries

In the Table 1 related experiment if it is available the photographs of the experiment (after 15days cd stress) it is better to add here. It could be better demonstration to readers.

Line no:60 to be corrected to “selenite and selenate” instead of selenite and selenite.

In the results authors used many times “significant (p < 0.05)”. Here better to use use significant without “(p < 0.05)” unless other value than (p < 0.05).

Please mentioned the details at first time use later use for short forms for selenite [Se (IV)] and selenate [Se (VI)].

Author Response

We are highly grateful for the helpful comments from the editor and referees that have been considered and carefully incorporated into the revised version of the manuscript. Please see our point-by-point response to the editor/reviewer’s comments.

Response to Reviewers’ Comments

Reviewer #2:

Comments and Suggestions for Authors

Xu et al manuscript entitled as “Effect of different forms of selenium on the physiological response and the cadmium uptake by rice under cadmium stress” studied the effect different form of Se on Cd stressed conditions in hydroponically grown rice seedlings. And authors concluded Se Met shown efficient effect on alleviating Cd stress in rice seedling than that other forms of Se.

Thank you very much for your valuable suggestions and comments.

Please find the some of the queries

In the Table 1 related experiment if it is available the photographs of the experiment (after 15days cd stress) it is better to add here. It could be better demonstration to readers.

Thank you very much for this valuable advice. Unfortunately, we had not taken photographs of the experiment. 

Line no:60 to be corrected to “selenite and selenate” instead of selenite and selenite.

We have made this correction in the revised version of the manuscript.

In the results authors used many times “significant (p < 0.05)”. Here better to use significant without “(p < 0.05)” unless other value than (p < 0.05).

Thank you. We have avoided the repetition of “p < 0.05” in the revised version of the manuscript.

Please mentioned the details at first time use later use for short forms for selenite [Se (IV)] and selenate [Se (VI)].

Thank you for this suggestion. Accordingly, we have introduced the abbreviations in the first instance and, after that, used the abbreviations alone. Please find the revision in the revised version of the manuscript.

We hope that the reviewer finds the revisions satisfactory.

Round 2

Reviewer 2 Report

Authors are improved the manuscript by considering the reviewer comments.

It was unfortunate, authors did not had experimental phonographs for the cd stress.